# Using Social Networks to Estimate the Number of COVID-19 Cases: The Incident (Hidden COVID-19 Cases Network Estimation) Study Protocol

**DOI:** 10.3390/ijerph18115713

**Published:** 2021-05-26

**Authors:** Honoria Ocagli, Danila Azzolina, Giulia Lorenzoni, Silvia Gallipoli, Matteo Martinato, Aslihan S. Acar, Paola Berchialla, Dario Gregori

**Affiliations:** 1Unit of Biostatistics, Epidemiology and Public Health, Department of Cardiac, Thoracic, Vascular Sciences, and Public Health, University of Padova, Via Loredan, 18, 35121 Padova, Italy; honoria.ocagli@unipd.it (H.O.); danila.azzolina@uniupo.it (D.A.); giulia.lorenzoni@unipd.it (G.L.); matteo.martinato@unipd.it (M.M.); 2Research Support Unit, Department of Translational Medicine, University of Eastern Piedmont, Via Solaroli, 17, 28100 Novara, Italy; 3Zeta Research, Via Antonio Caccia, 8, 34129 Trieste, Italy; silviagallipoli@zetaresearch.com; 4Department of Actuarial Sciences, Hacettepe University, 06800 Ankara, Turkey; aslihans@hacettepe.edu.tr; 5Department of Clinical and Biological Sciences, University of Torino, Regione Gonzole 10, 10043 Orbassano, Italy; paola.berchialla@unito.it

**Keywords:** COVID-19, NSUM, network scale-up method, hidden population

## Abstract

Recent literature has reported a high percentage of asymptomatic or paucisymptomatic cases in subjects with COVID-19 infection. This proportion can be difficult to quantify; therefore, it constitutes a hidden population. This study aims to develop a proof-of-concept method for estimating the number of undocumented infections of COVID-19. This is the protocol for the INCIDENT (Hidden COVID-19 Cases Network Estimation) study, an online, cross-sectional survey with snowball sampling based on the network scale-up method (NSUM). The original personal network size estimation method was based on a fixed-effects maximum likelihood estimator. We propose an extension of previous Bayesian estimation methods to estimate the unknown network size using the Markov chain Monte Carlo algorithm. On 6 May 2020, 1963 questionnaires were collected, 1703 were completed except for the random questions, and 1652 were completed in all three sections. The algorithm was initialized at the first iteration and applied to the whole dataset. Knowing the number of asymptomatic COVID-19 cases is extremely important for reducing the spread of the virus. Our approach reduces the number of questions posed. This allows us to speed up the completion of the questionnaire with a subsequent reduction in the nonresponse rate.

## 1. Introduction

Since December 2019, China and subsequently the whole world have been dealing with a pandemic due to a betacoronavirus related to the Middle East respiratory syndrome virus (MERS-CoV) and the severe acute respiratory syndrome virus (SARS-CoV2), named COVID-19 by the World Health Organization (WHO) [1]. The virus quickly spread globally [2,3]. In the Italian territory, the outbreak started with cases of pneumonia of unknown etiology at the end of January 2020.

Recent literature has highlighted a high percentage of undocumented cases among COVID-19-infected subjects. Such cases are mostly asymptomatic or paucisymptomatic, as their lack or scarcity of symptoms does not reach the attention of the healthcare system. Undocumented cases have been found to expose a higher proportion of the population due to the lack of quarantine measures [4] and to be hard to recognize, as asymptomatic or mildly symptomatic patients often do not seek medical attention due to a lack of symptoms [5]. While challenging, the prevalence estimation for asymptomatic and mildly symptomatic cases is very important given the highly contagious nature of the virus. Zou et al. [6] reported that the viral load in asymptomatic patients was similar to that in symptomatic carriers. Therefore, both asymptomatic and symptomatic patients may have the same transmissibility potential. Confirmed positive but asymptomatic people also need to be isolated to limit the contact with others. Consequently, accurate epidemiological monitoring of COVID-19 prevalence in asymptomatic people may also further decrease viral contagion. Moreover, it will help in the proper distribution of resources, tailoring the prevention program to the outbreak’s containment [7].

Several studies have tried to reveal undocumented cases. For example, using a networked dynamic metapopulation model and a Bayesian inference in mobility data within China, Li et al. [3] estimated that 86% of all infections were undocumented (95% CI: 82–90%) before the 23 January 2020 travel restrictions. Mizumoto et al. [8], in their study conducted on the Diamond Princess cruise ship, showed that there was a considerable proportion of asymptomatic individuals among all infected cases, which was 17.9% (95% credible interval: 15.5–20.2%). Other estimates of undiagnosed patients with COVID were among the evacuated citizens. Nishiura et al. [5,7] estimated a proportion of 33.3% asymptomatic cases (95% CI: 8.3–58.3%) among Japanese citizens evacuated from Wuhan. Undocumented infections seem to facilitate the geographic spread of SARS-COVID-19 [3]. In the Veneto region, in the municipality of Vo’, which was one of the initial outbreak sites in Italy, the choice to test the overall population helped to identify the proportion of positive COVID-19, revealing 37.7% asymptomatic patients (95% CI 25.5–51.9%) [9]. Properly estimating the number of COVID-19 positive cases, even if asymptomatic, is important since person-to-person transmission can occur from asymptomatic COVID-19 cases to the community, as shown in previous studies [10,11].

Traditional (i.e., direct) methods to detect positive cases are based on the seroepidemiological testing procedures of potentially exposed or infected populations. Real-time PCR tests or other laboratory tests may identify asymptomatic infections [12]. However, these approaches are time-consuming and require considerable financial resources [13]. Other indirect sample estimation methods, instead, are not suitable since they feature limitations, such as the use of independent samples, direct access to the source of data, or data of each country, as in capture-recapture technique, multiplier method, synthetic estimation and multivariate indicator methods, respectively [14,15].

Since undocumented infections are undefined, they constitute a hidden population. The network scale-up method (NSUM), which was first proposed by Bernard et al. [16,17], is among the recommended methods to estimate the hidden population available in the literature. NSUM has been widely used to estimate the size of hard-to-reach populations due to the stigmatizing nature of its knowledge, such as HIV [18], injuries [19], men who have sex with men (MSM) [20], and others. It relies on the idea that the probability of knowing someone in a specific subpopulation is related to the relative size of that subpopulation, i.e., the proportion computed based on the population size of all individuals.

## 2. Materials and Methods

The Hidden COVID-19 Cases Network Estimation (INCIDENT) study aims to develop a proof-of-concept study for estimating the number of undocumented COVID-19 infections using a Bayesian approach of the traditional NSUM.

### 2.1. Study Design

This is a cross-sectional survey-design study to assess the prevalence of undocumented COVID-19 symptoms in Italy using an anonymous online questionnaire. The data were collected starting on 15 April 2020 and will continue until the winter of 2020.

#### 2.1.1. Procedures

To avoid unnecessary interactions, this study was structured only for electronic distribution given the social distancing and the limitations imposed by the Italian government.

The questionnaire was created through LimeSurvey (LimeSurvey GmbH, Hamburg, Germany), a professional open-source online survey tool. The respondents had to be at least 16 years old and were required to sign an informed consent.

The study design mimicked snowball social network sampling but had a nonrandom entry point in the population. The snowball sampling method was chosen since it is widely applied and evaluated as particularly useful in studies that consider hidden populations as a target [21,22]. The questionnaire was advertised via social networks, mobile messaging systems, emails, and newspapers (Appendix A).

#### 2.1.2. Ethics

The INCIDENT study was approved by the Ethics Committee of the University of Torino, protocol No 458163, 10 October 2020.

### 2.2. The Network Scale-Up Method in the Literature

NSUM was first applied to estimate the death toll of the 1985 Mexico City earthquake [17] due to the missing reports regarding fatalities by official registry. Since then, NSUM has been widely used to estimate the size of a subpopulation that consists of hard-to-identify individuals [23], such as individuals with high-risk behaviors that lead to stigmatization and discrimination, such as individuals living with HIV/AIDS [24,25,26,27,28,29,30,31], MSM [20,32], sex workers [23,33], drug addicts [33,34,35,36,37,38], or alcohol users [34,39]. NSUM has also been used to estimate the number of treatment failures [40], people with disabilities [41], number of abortions [42,43], and suicide attempts [43]. Teo et al. [26] suggested the use of NSUM for estimating the hidden population to improve surveillance, prevention, and treatment after proper methodological adjustments.

Table 1 shows articles using NSUM to estimate hidden populations. The search was based on a review in PubMed (Figure 1).

In the Italian context, NSUM has been used to estimate the number of children with foreign body injuries [44] and to assess the perceived quality of care (PQC) in an oncological center. In the latter study, estimation of PQC was lower than in the traditional questionnaire; in some cases, the level of dissatisfaction was 20-fold higher [45]. Among other selected applications, Paniotto et al. [46] estimated the number of drug addicts, sex workers, and MSM in their study, showing that NSUM estimates were similar or lower compared to other estimation methods. Additionally, estimated populations of seropositive, homeless, and female victims of violence in Killworth et al. [24] were comparable to official data. One limitation of NSUM is that social and physical barriers, such as ethnicity, occupation, or location of residence, may influence the likelihood that respondents know people in hidden populations. This is known as the barrier effect. On the other hand, individuals may not know everything about other people in their personal network. This instance, in which a contact does not share information with the respondent, is termed transmission bias. Other limitations in applying these methods include recall bias [29] and response bias [47]. Several authors have tried to address such limitations, as shown in Table 1. McCormick et al. [48] adjusted NSUM for recall bias, various authors adjusted for barrier effects [49] and transmission errors [28,29,30,39,42,48,50], and Jing et al. [23] adjusted for response bias.

#### 2.2.1. NSUM Questionnaire

This questionnaire is structured into three sections: (1) four questions for the demographic characteristics of the respondent (gender, age, nationality, and region of residence), (2) four target questions related to COVID-19 disease that were defined by consulting the available literature on COVID-19 [61], and (3) one question used for the estimation of social network size that was randomly drawn from 15 known populations (see Appendix A for the full report of the questionnaire). The data source for the known population size is the Italian National Institute of Statistics (ISTAT) [62] (Table 2).

All questions concerning the specific subpopulations were introduced with the sentence: “How many people do you know …?”. In this study, we used the definition of knowing someone based on those provided by Bernard et al. [63] and already used in previous Italian studies [44,45].

#### 2.2.2. NSUM Assumption

The NSUM estimation method, as explained by Bernard et al. [63] “rests on the assumption that people’s social networks are, on average, representative of the general population in which they live and move.” For example, if a responder assesses to knowing 100 subjects on average and two of them are COVID-19 positive, we estimate a prevalence of 2/100 COVID-19-positive subjects considering as reference point the personal network size. This estimated prevalence is combined with the known size of the general population to estimate the size of hard-to-reach populations, such as the COVID-19-infected population. The accuracy of the estimated size of the hidden population increases as the number of people who answer the question increases.

The NSUM, however, has some limitations. For example, people may not know all the characteristics of their personal network (i.e., a respondent may not know that a member of his or her network is affected by COVID-19.) This is called the transmission bias [48]. In addition, social and physical barriers, such as ethnicity, race, occupation, and location of residence, can cause variations in the probability that respondents know people in hidden populations; this is called the barrier effect [24]. Despite these biases, NSUM has two major advantages. First of all, this method does not ask the respondent for information on its characteristics. For example, stigmatized or hidden populations may be reluctant to disclose their status even in an anonymous survey [51]. Secondly, it is not necessary to directly interview the members of a hidden population, but the NSUM allows the use of considerably cheaper and easier-to-implement sampling techniques that make use of established sampling frames [38].

#### 2.2.3. NSUM Estimator

Letter *T* is the size of the general population, $m_{ik}$ is the number of subjects in the hidden population known by individual *i* in the subpopulation, *k* and ${\hat{c}}_{i}$ is the estimated average size of the social network related to the individual *i.* The scale-up estimator is based on the assumption that the number of subjects known to the respondent in the *k*-th subpopulation follows a binomial distribution [64] where $m_{ik}\sim Binom\left(c_{i},\frac{T_{k}}{T}\right)$. The scale-up estimator of the hidden population size is obtained by the following equation:(1)$${\hat{e}}_{k}=T\frac{\sum_{i}m_{ik}}{\sum_{i}{\hat{c}}_{i}}$$

To estimate the size of the hidden population, we follow three steps:(1)Estimate the average size of the personal network, $c_{i}$, by asking how many people the respondent personally knows about the *k* known populations (e.g., the number of people who were married in 2019). This number will then be divided by the number of people who got married in 2019 in Italy ($T_{k}$), where $T_{k}$ is the total size of subpopulation *k*.
(2)$${\hat{c}}_{i}=T\frac{\sum_{k=1}^{K-1}m_{ik}}{\sum_{k=1}^{K-1}T_{k}}$$(2)Define the number of hidden COVID-19 cases present in each social network, for example, by asking the respondent how many people he/she knows with COVID-19.(3)Calculate the COVID-19 population size obtained by multiplying the estimated proportions of the population in each subpopulation by the general Italian population. For example, if a respondent knows 10 subjects with COVID-19 cases and has a personal network of 100 people and the total population is 1,000,000, the estimated number of hidden COVID-19 cases will be approximately calculated as: 10/100 × 1,000,000.


### 2.3. Bayesian NSUM Estimation

Under maximum likelihood estimation, several known populations should be used to reduce the variance of the estimates. This prolongs the time required to complete the questionnaire by increasing the likelihood of dropouts and nonresponses. Estimating the size of the network by considering the known population as partially unknown could be a solution to shorten the length of the survey. For this reason, the Bayesian estimation methods proposed by Maltiel et al. [29] will be extended to estimate the unknown network size using the Markov chain Monte Carlo (MCMC) algorithm.

Moreover, the original personal network size-estimation method proposed by Killworth and colleagues [64] was based on fixed-effects, maximum likelihood estimators in which the network size was considered a nonrandom component. Instead, Maltiel et al. [29] extended this approach in a Bayesian setting by treating personal network sizes as random variables. This allows us to generalize the model to account for the variation in respondents’ propensity to know people in particular subgroups (barrier effects), such as their tendency to know people like themselves, and their lack of awareness to recognize their contacts’ group memberships (transmission bias).

#### 2.3.1. Extended Random Degree Model

The NSUM formulation proposed by Maltiel et al. [29] assumes that the estimate of an individual’s network degree, $c_{i}$, improves if a respondent knows a considerable number of subjects in a subpopulation. Network estimation is embedded into a Bayesian hierarchical modeling framework where the lognormal distribution best fits the network estimates across multiple datasets [29].
(3)$$\begin{array}{c} m_{ik}\sim Binom\left(c_{i},\frac{T_{k}}{T}\right)\\ c_{i}\sim \log\;\mathrm{Normal}\;\left(\mu ,\sigma^{2}\right) \end{array}$$

$m_{ik}$ values are the number of subjects that the *i*-th subject knew in the *k*-th subpopulation. Observed $m_{ik}$ values are assumed to be a realization of a binomial random variable whose parameters are defined by the personal network degree ($c_{i}$) and the overall known proportion of subjects in the subpopulations ($\frac{T_{k}}{T}$), where $T_{k}$ is the size of the *k*-th known sub-population and *T* is the overall population size.

The parameters of the random degree will be estimated in a Bayesian manner using the uninformative priors $\pi \left(T_{K}\right)$ for the *k*-th subpopulation, as in the Maltiel et al. [29] work:(4)$$\begin{array}{c} \pi \left(T_{K}\right)\propto \frac{1}{T_{K}}1_{\;T_{K}}\leq T\\ \mu \sim U\left(3,8\right)\\ \sigma \sim U\left(\frac{1}{4},2\right) \end{array}$$

$T_{K}$ priors (hidden population parameter) have been used in the literature for Bayesian estimation of population size with vague prior [29]. The *μ* and *σ* prior distributions were derived by Maltiel et al. [29], identifying the best fit to the network estimates across multiple datasets.

In our NSUM formulation, the number of subjects known by the respondent in the *k*-th subpopulation is unknown, except for the target question that identifies the hard-to-reach subpopulation and the question drawn of the known subpopulation extracted for each respondent from the list of the known subpopulation.

The model is reformulated assuming $m_{ik}$ is partially unknown. The number of subjects that the *i*-th respondent knows in the *k*-th subpopulation for each MCMC iteration will be drawn from a binomial random variable ($m_{ik}$) except for the target question that identifies the hard-to-reach subpopulation and the known population.

#### 2.3.2. Performance of the Modified Maltiel Estimators

The algorithm was initialized at the first iteration from the first resampled $m_{ik}$ values defining the starting values using the Killworth method [64].

The performance of the proposed NSUM estimator was evaluated in a simulation study considering different study size scenarios (1000, 1500, 2000, 2500, 3000).

Answers were generated by assuming the sizes of known subpopulations as indicated in the McCarty NSUM study [64] for each scenario. Data were generated using the original nsum.simulate() function proposed by McCarty in the NSUM R package [65]. The data were generated 300 times. For each replication, Maltiel’s NSUM model and our NSUM proposal were estimated with 500 iterations, discarding the first 50 (burn-in iterations).

The total Italian population in January 2020 was 60,244,639 [62]. The assumed prevalence of COVID-19-positive cases is 1.37% (800,000 cases, like the peak of 805,947 officially diagnosed positive cases on 22 November 2020).

To compute the modified version of Maltiel’s short-form questionnaire for each responder, we randomly sampled one of the responses characterizing the known populations from the full generated database, leaving the other responses as missing.

The 95% credible interval (CI) was computed.

The performance of the estimators was computed by calculating the average CI over simulations with the average prevalence estimate and the bias for each study size.

Computations were performed using R 3.6.2 [66] software.

### 2.4. Statistical Analysis

Data will be presented in aggregate form, and it will not be possible to trace information or make comparisons on an individual level.

Continuous variables will be summarized using the median (quartiles I and III), and qualitative variables will be synthesized using percentages and absolute numbers. Comparisons between groups will be identified using the Wilcoxon–Kruskal–Wallis test for continuous variables and the Pearson chi-square test for categorical variables.

## 3. Results of the Simulation Study

The INCIDENT study is still ongoing. A total of 1963 questionnaires have been collected until 6 May 2020, 1703 were completed except for the random questions, and 1652 were completed in all three sections. The respondents were primarily female (1206, 61%), and the prevalent residents were the areas most affected by COVID-19, including Veneto, Lombardia, and Piemonte.

### Performances of the Modified Maltiel’s Estimators

The estimated prevalence remains approximately constant as the sample size increases for both Maltiel’s method (close to 1.37%) and the modified version (close to 1.57%). The bias remains approximately −0.05% and 0.2% for the original version and the modified version of the method, respectively. The length of the interval decreases as the sample size increases. A sample size of 2000 subjects guarantees a CI length of 0.044% for the modified method (bias = 1.197) and 0.038 (bias = −0.051) for the original approach (Table 3).

The modified version of the method demonstrates a slightly higher bias than the original method; however, prevalence estimates are similar between the two methods.

## 4. Discussion

The results of the simulation study proposed in this protocol revealed that our NSUM model demonstrates a slightly higher bias than the original method. However, prevalence estimates are similar between the two methods.

The NSUM as modified in this study has numerous advantages. First, questions for the scale-up social network estimator can be easily integrated into the survey. As shown in our literature review, most of the time needed to complete the questionnaire is used to estimate the size of the social network [26,28,30,32,40,43,44,45,51]. Some authors have preferred to estimate the size of the active network population only once, as in a previous Iranian population study [60], in order to reduce the time for survey completion.

In our study, we posed questions only related to the hidden population [31,35,36,37,39,41,42,54,57]. Few studies have chosen to use questions related to known populations [17,20,25,26]. Ahmadi et al. [34] proposed only one known population, as in our work. However, with traditional NSUM, our questionnaire would have needed 15–20 known populations; with our method, only one question is needed. This allows to speed up the completion of the questionnaire by reducing the dropout and nonresponse rate.

Moreover, the modified NSU method allows estimation of the size of the personal network independently from the survey on COVID-19. This is possible because data for the estimation of the size of the social network are collected on a separate subpopulation from the general survey.

Direct methods for the estimation of undocumented cases are resource and time consuming due to the requirement of large-scale testing procedures. Viable alternatives that may overcome such limitation are the introduction of statistically strong sampling design for estimating the parameters of the epidemics, such as the one proposed by Alleva et al. [67]. Other methodological statistical studies have tried to estimate the true number of COVID-19-infected people indirectly. Palatella et al. [4] tried to estimate the number of cases based on PCR test alone and Noh and Danuser [68] based on government daily counts of confirmed cases. However, these estimates could underestimate the real proportion since the data are related to the specific population.

In literature, there are different indirect sample estimation methods, but those are not suitable for the purpose of our study. For example, capture-recapture technique requires at least two independent and representative samples [14]. Multiplier and enumeration methods, on the other hand, require direct access to the target population [15].

## 5. Conclusions

NSUM shows its advantages when we utilized it for the same survey in different contexts, as in our study with snowball sampling. Therefore, data for the estimation of the social network size can be collected in a standardized way both spatially and temporally. This would be extremely difficult if the methods instead required direct access to the hidden population as the target of the investigation. Despite the bias of the traditional NSUM, our modified version shows advantages, considering that only one question was used for defining the social network.

The NSUM can help tracing the undocumented cases that could be known in the people’s social network but do not draw the attention of the healthcare system. Our method would be useful when government testing is not widespread enough to reach the overall population, such as with COVID-19.

## Figures and Tables

**Figure 1 ijerph-18-05713-f001:**
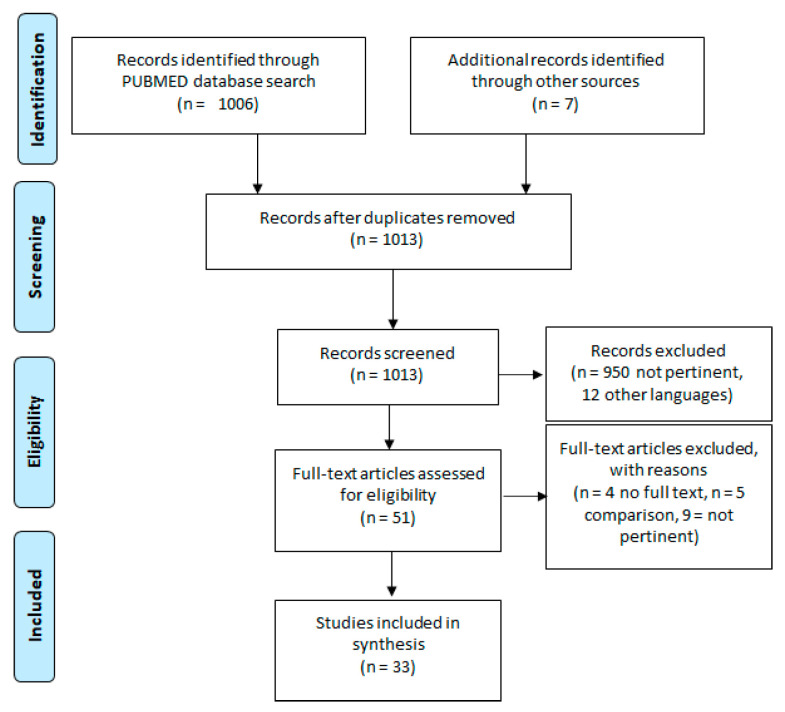
Flowchart study review.

**Table 1 ijerph-18-05713-t001:** Studies that estimate the hidden population using the network scale-up method or its modification.

Author	Year *	*N* Known Population	Hidden Population	*N* Respondents	Method	Adjustment	Place
Ahmadi [34]	2019	1	Drug/alcohol users before driving	363	NSUM		Iran
Bernard [17]	1991	6	Deaths in earthquake		NSUM		Mexico
Carletti [45]	2017	20	Oncological patients	299	NSUM		Italy
Ezoe [20]	2012	3	Men who have sex with men		NSUM	Transmission Error	Japan
Feehan [27]	2016	22	Populations at risk for HIV/AIDS	4669	Blended Scale-up		Rwanda
Guo [25]	2013	3	Populations at risk for HIV/AIDS	2957	NSUM		China
Habecker [51]	2015	18	Moved to Nebraska in US during last 2 years, do not approve of interracial dating, heroin users	618	Mean Of Sums NSUM		United States
Haghdoost [52]	2015	**	Population of breast, ovarian/cervical, prostate, and bladder cancers	3052	NSUM		Iran
Heydari [40]	2019	25	Treatment failure	2550	NSUM		Iran
Jafari [28]	2014	29	Populations at risk for HIV/AIDS	500	NSUM	Transmission Bias, Barrier Effects	Iran
Jing [23]	2018	48	Female sex worker		RRT, NSUM	Response Bias	China
Kadushin [38]	2006	3	Heroin users		NSUM		United States
Kazemzadeh [53]	2016	**	High-risk behaviors	563	CM, NSUM		Iran
Killworth [24]	1998	24	HIV prevalence, women who have been raped, the homeless	1554	NSUM		United States
Maghsoudi [33]	2014	20	Injection drug users, female sex workers	600	NSUM	Barrier Effect	Iran
Maltiel [29]	2015	29	Populations at risk for HIV/AIDS	500	Bayesian NSUM	Transmission Bias, Barrier Effects	Brazil
Mccormick [48]	2010	12	Personal network size	1370	Latent Non-Random Mixing Model NSUM	Transmission Bias, Barrier Effects, And Recall Bias.	Brazil
Mohebbi [41]	2014	**	People with disabilities	3052	NSUM		Iran
Moradinazar [54]	2019	**	Suicides and suicide attempts	500	NSUM		Iran
Narouee [35]	2019	**	Injection drug users	1000	NSUM	Barrier Effect	Iran
Narouee [55]	2020		Rural area	1000	MLE—NSUM		
Nikfarjam [39]	2017	**	Alcohol use	12,000	NSUM	Transmission Bias, Barrier Effects	Iran
Nikfarjam [36]	2016	**	Illicit drug users	7535	NSUM		Iran
Rastegari [42]	2014	**	Abortions	12,960	NSUM	Transmission Bias, Barrier Effects	Iran
Sajjadi [50]	2018	6	Students with high-risk behaviors	801	NSUM	Transmission Bias, Barrier Effects	Iran
Salganik [30]	2011	20	Populations at risk for HIV/AIDS		NSUM, GSU	Transmission Bias, Barrier Effects	Brazil
Shokoohi [56]	2010	6	Network	500	NSUM		Iran
Shokoohi [31]	2012	**	Populations at risk for HIV/AIDS	500	NSUM		Iran
Snidero [44]	2012	33	Foreign body injuries	1081	NSUM		Italy
Teo [26]	2019	24	Populations at risk for HIV/AIDS	199	Bayesian NSUM		Singapore
Vardanjani [57]	2015	**	Cancer	195	Generalized NSUM		Iran
Wang [32]	2015	22	Men who have sex with men	3097	NSUM		China
Zahedi [37]	2018	**	Drug users	2157	NSUM	Barrier Effect	Iran
Zamanian [58]	2016	25	Age-gender distribution of women	1275	NSUM		Iran
Zamanian [59]	2019	25	Abortion	1500	NSUM	Barrier Effect	Iran

*N*, Number; MLE, maximum likelihood estimation; NSUM = network scale-up method; RRT = randomized response technique; CM, crosswise model. * year of publication, ** Rastegari et al. [60].

**Table 2 ijerph-18-05713-t002:** List of known-size populations used in the questionnaire for the estimation of respondents’ network sizes.

Sub-Population of Known Size	Population Size	Reference Year	Source
People who separated	99,611	2016	Demographic model
Foreign residents	5,255,503 ***	2019	Demographic model
Victims of car accidents with injuries	3334	2018	Demographic model
People who graduated	8530 **	2018	MIUR
People working part-time	3,689,153 ***	2019	Demographic model
Three-member families	4954	2019	AVQ
Cohabiting couples	14,110	2019	AVQ
People who married	195,778	2018	Demographic model
Children born	440,780	2018	Demographic model
People above 14 with smoking habits	10,122	2017	AVQ
People using the mass media (newspapers, magazines, TV, radio, etc.)	86,142 *	2017	AVQ
People who attend places of worship	14,264	2018	AVQ
People who walk to work	2750	2018	AVQ
People who go to school by bus	8743	2018	AVQ
Three-year-olds and above who used a PC and Internet	62,232	2017	AVQ

* Number watching TV 53953, radio 32189. ** http://dati.ustat.miur.it/dataset/laureati/resource/43df861d-7345-481a-9803-2eb236aa022e (accessed on 16 April 2020), (difference between 2018 and 2017, 326332-317802). *** cumulative data. MIUR, Ministry of Instruction, University and Research, http://dati.ustat.miur.it/dataset/laureati (accessed on 16 April 2020); AVQ, Aspects of daily life survey, http://dati.istat.it/ (accessed on 16 April 2020).

**Table 3 ijerph-18-05713-t003:** Simulation results for the modified Maltiel’s method performance.

Study Size	Maltiel’s Method				Modified Maltiel’s Method		
	Benchmark Prevalence	Prevalence%	95% CI Length	Bias	Prevalence%	95% CI Length	Bias
1000	1.37	1.324	0.056	−0.056	1.561	0.074	0.191
1500	1.37	1.327	0.044	−0.053	1.57	0.056	0.200
2000	1.37	1.329	0.038	−0.051	1.567	0.044	0.197
2500	1.37	1.329	0.031	−0.051	1.566	0.038	0.196
3000	1.37	1.33	0.025	−0.05	1.574	0.032	0.204

For each study size, the average prevalence (%) and the 95% CI were reported for Maltiel’s method and the modified version. The benchmark prevalence represents the true value used to generate the data. The bias (average estimated prevalence—benchmark) was reported for each method.

## Data Availability

The data presented in this study are available on request from the corresponding author. The data are not publicly available due to privacy.

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
