# Peer review of "Using Social Networks to Estimate the Number of COVID-19 Cases: The Incident (Hidden COVID-19 Cases Network Estimation) Study Protocol"

_ijerph, 2021, doi:10.3390/ijerph18115713_

Round 1

Reviewer 1 Report

Comments on the manuscript

Using social networks to estimate the number of COVID-19 cases:

The INCIDENT (hIddeN CovID-19 casEs Network estimation) study protocol

Manuscript ID: ijerph-1153833

The submitted manuscript is the study protocol of an ongoing study. Although it seems the study is already in an advanced stage, I have major concerns about the validity of the promised results.

Main remarks

Estimation of the proportion of asymptomatic cases

The introduction emphasizes the importance of estimating the proportion of “asymptomatic or mildly symptomatic” Covid 19 cases in an exposed population. This task is extremely challenging. However, Section 2 scales down this aim to the problem of estimating the proportion of “undocumented cases”. This seems a more realistic but quite different aim.

A proof-concept study using a Bayesian approach to the network scale-up method (NSUM) is described. In general, NSUM can be seen as a “cheap and easy” method to estimate the size of a hidden population. It is based on questioning (a random sample of) social network users about the health conditions of their acquaintances. Unfortunately it relies on very doubtful assumptions (such as “people’s social networks are, on average, representative of the general population in which they live and move” or “everyone knows very well the behaviours of their acquaintances”, etc.). Additional technical assumptions about the distributions of key quantities (such as mik ci, …) are also required for estimation purposes. The plausibility of these assumptions has to be assessed in each particular study. Unfortunately, it is absolutely not questioned by the authors of this paper. Any kind of evaluation of the impact of these assumptions on the results is not taken into account. The fact that “NSUM has been widely used to estimate the size of hard-to-reach populations” (a list of 35 studies is given in the paper) is no guarantee that it would provide reliable estimates of the proportion of undocumented cases and, a fortiori, of the asymptomatic Covid 19 cases.

As a statistician, I would have many reservations about the use of such a method for the purpose of estimating the number of asymptomatic cases. To my knowledge, honest estimates, not relying on doubtful assumptions, can only be obtained by testing samples of subjects randomly taken from the exposed population. Efficient methods for this purpose have been proposed (e.g., [1]). Usable data are also provided by the “mass testing” programmes recently implemented in different countries. Of course, these procedures are more expensive than questioning the users of a social network, relying on their knowledge of the behaviours of their acquaintances and the correctness of their answers.

Simulation study

The technical part of the manuscript describes the statistical model (Matiel’model) and the estimation procedure that are planned for the data analysis. A small simulation study illustrates the performance of the estimation procedure.

The quality of the description is poor (see the minor remarks below). It is not clear whether there is some methodological originality (is the use of Markov chain Monte Carlo a new idea in this framework?). The simulation shows that the estimation procedure provides reasonable results assuming that the model is correct (although the bias does not decrease with increasing study size). However, for the reasons mentioned above, this is no promise of reliable estimates of the real target proportion.

Some minor remarks

Line 85: “electronic dispersion”: is this correct? (electronic distribution?)

Section 2.1.1: is the questionnaire available? Where?

Line 90: is the non-random entry point not a problem for statistical inference?

Line 135: demographic characteristics of the respondent?

Line 139: “… 15 known populations shown in Table 2? To avoid confusions, the word “population” should only be used for the target population (e.g., Italy); the words “sub-population” or “auxiliary population” would seem more appropriate for the populations of Table 2.

Line 151: “… to estimate the size of the hard to reach population”.

Line 156: Tk and ci are still undefined at this point (size of the k-th known sub-population; size of the of the social network of user i).

Line 159: mik id the number of subjects in the hidden population.

Line 162: “asking how many people the respondent personally knows who belong to the k-th known sub-population (e.g., the number of people who married in 2019)”.

Line 165: K is undefined in the formula (number of known sub-populations).

Line 167: A case “with flu symptoms” is not equivalent to an “asymptomatic case” (or even to an “undocumented” or a “hidden Covid case”). This seems a major flaw.

Step (3). In my opinion, only the “flu population size” is estimated by means of the described procedure (as long as “flu” has been exactly defined).

Line 200: Is “network degree” a synonym of “size of the personal network”?

Lines 200-202: As far as I understand, the “known fixed population sizes” are replaced by “random subpopulation sizes”. Unfortunately, the formula defining the prior π(Tk) is completely unclear: N and 1T are undefined.

Lines 210-2011: “the question drawn of the known subpopulation extracted … from the list of the known subpopulation”??

Lines 236-241: These lines refer to elementary wishes for the final report and are unnecessary in a manuscript submitted for publication.

Line 243. The title “3. Results” is misleading. The results of the study are not presented in the study protocol.

Sections 2.3.1 and Section 3.1 describe “A simulation study” and the “Results of the simulation study”.

Line 244: Questionnaires have been collected in the year 1963?

Reference

[1] Alleva G., Arbia G., Falorso P.D. (2020) A sample approach to the estimation of the critical parameters of the SARS-CoV-2 epidemics: an operational design with a focus on the Italian health system DOI: 10.13140/RG.2.2.25507.40481

Author Response

Comments on the manuscript

Using social networks to estimate the number of COVID-19 cases:

The INCIDENT (hIddeN CovID-19 casEs Network estimation) study protocol

Manuscript ID: ijerph-1153833

The submitted manuscript is the study protocol of an ongoing study. Although it seems the study is already in an advanced stage, I have major concerns about the validity of the promised results.

Main remarks

Estimation of the proportion of asymptomatic cases

The introduction emphasizes the importance of estimating the proportion of “asymptomatic or mildly symptomatic” Covid 19 cases in an exposed population. This task is extremely challenging. However, Section 2 scales down this aim to the problem of estimating the proportion of “undocumented cases”. This seems a more realistic but quite different aim.

Thank you for pointing this out. The introduction has been modified.

A proof-concept study using a Bayesian approach to the network scale-up method (NSUM) is described. In general, NSUM can be seen as a “cheap and easy” method to estimate the size of a hidden population. It is based on questioning (a random sample of) social network users about the health conditions of their acquaintances. Unfortunately it relies on very doubtful assumptions (such as “people’s social networks are, on average, representative of the general population in which they live and move” or “everyone knows very well the behaviours of their acquaintances”, etc.). Additional technical assumptions about the distributions of key quantities (such as mik ci, …) are also required for estimation purposes. The plausibility of these assumptions has to be assessed in each particular study. Unfortunately, it is absolutely not questioned by the authors of this paper. Any kind of evaluation of the impact of these assumptions on the results is not taken into account. The fact that “NSUM has been widely used to estimate the size of hard-to-reach populations” (a list of 35 studies is given in the paper) is no guarantee that it would provide reliable estimates of the proportion of undocumented cases and, a fortiori, of the asymptomatic Covid 19 cases.

Thanks for pointing this out, your insights are both justified and useful. Assumptions such as the generalisability of the personal network size to estimate the prevalence of the phenomenon in the general population is an underlying assumption of the NSUM method. However, as indicated in section 2.2.2, (lines 272-283) this method has two major advantages. First, the NSUM methods do not ask the respondent for information on respondent’s characteristics. For example, stigmatised or hidden populations may be reluctant to disclose their status even in an anonymous survey. Secondly, it is not necessary to directly interview members of a hidden population, since the NSUM allows the use of considerably cheaper and easier to implement sampling techniques that make use of established sampling frames.

On the other hand, this is a study protocol that described a method according to the traditional NSUM assumptions and our modifications, further evaluation of the behaviour of the NSUM estimators will be carried out during data analysis phase.

As a statistician, I would have many reservations about the use of such a method for the purpose of estimating the number of asymptomatic cases. To my knowledge, honest estimates, not relying on doubtful assumptions, can only be obtained by testing samples of subjects randomly taken from the exposed population. Efficient methods for this purpose have been proposed (e.g., [1]). Usable data are also provided by the “mass testing” programmes recently implemented in different countries. Of course, these procedures are more expensive than questioning the users of a social network, relying on their knowledge of the behaviours of their acquaintances and the correctness of their answers.

The choice of the method depends on the research objective. The purpose of the study protocol was the identification of undocumented COVID-19 cases (lines 80-81) during the first wave of pandemic. This population group was not easy to reach using traditional sampling tools. Moreover, especially during the first months of the pandemic, due to the general uncertainty, declaring one's disease status could be stigmatising. Valid and well known estimates have been carried out in literature by using the NSUM, as the brief review in the point .2,, especially when targeting hard-to-reach populations. The alternative solution you correctly suggest is surely a viable option but, is much more time and resource consuming one than indirect methods.

Simulation study

The technical part of the manuscript describes the statistical model (Matiel’model) and the estimation procedure that are planned for the data analysis. A small simulation study illustrates the performance of the estimation procedure.

The quality of the description is poor (see the minor remarks below). It is not clear whether there is some methodological originality (is the use of Markov chain Monte Carlo a new idea in this framework?). The simulation shows that the estimation procedure provides reasonable results assuming that the model is correct (although the bias does not decrease with increasing study size). However, for the reasons mentioned above, this is no promise of reliable estimates of the real target proportion.

The use of MCMC is not a new element in this work. The approach was first proposed in NSUM by Maltiel et el. as reported in the paragraph 2.3. Bayesian NSUM estimation (lines 419-434).

The method proposed in this study proptocol is an extension of the Maltiel’s model. In our NSUM formulation (see paragraph 2.3.1 extendend random degree model), the number of subjects known by the respondent in the k-th subpopulation is unknown, except for the target question, that identifies the hard-to-reach subpopulation and the question extracted for each respondent, from the list of the known subpopulation. The number of subjects that the i-th respondent knows in the k-th subpopulation for each MCMC iteration, will be drawn from a binomial random variable except for the target question that identifies the hard-to-reach subpopulation and the randomly drawn known population.

This mechanism slightly increases the bias of the estimates compared to the original model formulation but allows the use of a smarter and more respondent-friendly survey instrument. We agree that the bias is not affected by sample size, however the variability of the estimates is strongly influenced by it. The proposed work is a study protocol, further insights into the behaviour of the estimators will be made in the final analysis work.

Some minor remarks

Line 85: “electronic dispersion”: is this correct? (electronic distribution?)

Thanks for pointing this out, modified accordingly.

Section 2.1.1: is the questionnaire available? Where?

The questionnaire is available in the supplementary material-

Line 90: is the non-random entry point not a problem for statistical inference?

Thanks for pointing this out. A non-random sampling strategy has been considered because the target population is difficult to reach and there is no potential participant list to draw from. The NSUM method has been validated on this type of sampling strategy as demonstrated in the literature.

Line 135: demographic characteristics of the respondent?

Yes, it is correct. Added to be clearer.

Line 139: “… 15 known populations shown in Table 2? To avoid confusions, the word “population” should only be used for the target population (e.g., Italy); the words “sub-population” or “auxiliary population” would seem more appropriate for the populations of Table 2.

Thanks for pointing this out. The text has been modified as suggest.

Line 151: “… to estimate the size of the hard to reach population”.

Thanks for the suggestion, modified accordingly.

Line 156: Tk and ci are still undefined at this point (size of the k-th known sub-population; size of the of the social network of user i).

Thanks for the suggestion, modified accordingly.

Line 159: mik id the number of subjects in the hidden population.

Thanks for the suggestion, modified accordingly.

Line 162: “asking how many people the respondent personally knows who belong to the k-th known sub-population (e.g., the number of people who married in 2019)”.

Thanks, modified as suggested.

Line 165: is undefined in the formula (number of known sub-populations).

Modified as suggested.

Line 167: A case “with flu symptoms” is not equivalent to an “asymptomatic case” (or even to an “undocumented” or a “hidden Covid case”). This seems a major flaw.

Thanks for pointing this out. We have modified in order to be clear.

Step (3). In my opinion, only the “flu population size” is estimated by means of the described procedure (as long as “flu” has been exactly defined).

Line 200: Is “network degree” a synonym of “size of the personal network”?

Yes, it is correct.

Lines 200-202: As far as I understand, the “known fixed population sizes” are replaced by “random subpopulation sizes”. Unfortunately, the formula defining the prior π(Tk) is completely unclear: and 1T are undefined.

Thanks, modified the manuscript.

Lines 210-2011: “the question drawn of the known subpopulation extracted … from the list of the known subpopulation”??

Thanks for the suggestion. Modified the text to be clearer (lines 471-474).

Lines 236-241: These lines refer to elementary wishes for the final report and are unnecessary in a manuscript submitted for publication.

Thanks, removed as suggested.

Line 243. The title “3. Results” is misleading. The results of the study are not presented in the study protocol.

Modified as suggested.

Sections 2.3.1 and Section 3.1 describe “A simulation study” and the “Results of the simulation study”.

Thanks for pointing this out, modified as suggest.

Line 244: Questionnaires have been collected in the year 1963?

No, 1963 refers to the number of the questionnaires collected until May 6, 2020. Modified in order to avoid the same misunderstanding.

Reference

[1] Alleva G., Arbia G., Falorso P.D. (2020) A sample approach to the estimation of the critical parameters of the SARS-CoV-2 epidemics: an operational design with a focus on the Italian health system DOI: 10.13140/RG.2.2.25507.40481

Submission Date

05 March 2021

Date of this review

22 Mar 2021 16:06:52

Reviewer 2 Report

The authors aim to develop a proof-of-concept method for estimating the number of undocumented infections of COVID-19 in Italy, using social networks. They propose a method based on a Bayesian approach of the network scale-up method (NSUM). They present performances of the modified Maltiel’s estimators as results of their ongoing project.

My comments on your work are:

1. In Figure 1, verify the number of articles evaluated for eligibility.
2. It is suggested to mention if there is another alternative to the NSUM and compare it with the method you propose.
3. You mention “All questions, except the first four, were introduced as follows: How many people do you know…?”. How many questions per section?
4. The equations are not numbered.
5. Verify the following statements: “burnin iterations…” (line 225), it should be burn-in, “In our NSU formulation…” (line 208), “A sample size of 2000 subjects guarantees a CI of 0.044% .. . ”(Line 254).
6. The discussion focuses on other aspects rather than the proposed method. This section needs to be improved.
7. Briefly mention what the next stages of your project would consist of.

Author Response

2nd reviewer

Open Review

English language and style

( ) Extensive editing of English language and style required
( ) Moderate English changes required
(x) English language and style are fine/minor spell check required
( ) I don't feel qualified to judge about the English language and style

Yes

Can be improved

Must be improved

Not applicable

Does the introduction provide sufficient background and include all relevant references?

(x)

( )

( )

( )

Is the research design appropriate?

(x)

( )

( )

( )

Are the methods adequately described?

(x)

( )

( )

( )

Are the results clearly presented?

( )

(x)

( )

( )

Are the conclusions supported by the results?

( )

(x)

( )

( )

Comments and Suggestions for Authors

The authors aim to develop a proof-of-concept method for estimating the number of undocumented infections of COVID-19 in Italy, using social networks. They propose a method based on a Bayesian approach of the network scale-up method (NSUM). They present performances of the modified Maltiel’s estimators as results of their ongoing project.

My comments on your work are:

  1. In Figure 1, verify the number of articles evaluated for eligibility.

Thanks, the figure has been modified.

  1. It is suggested to mention if there is another alternative to the NSUM and compare it with the method you propose.

Good point. The alternative, as reported in lines (50-53), is to use direct methods for the estimation of hidden population, which however are time consuming, or other indirect methods as reported in lines 53-56, which in this context are not suitable.

  1. You mention “All questions, except the first four, were introduced as follows: How many people do you know…?”. How many questions per section?

All questions concerning the specific subpopulations were introduced with the sen-tence: “How many people do you know…?”.  Questions are divided as follow for each respondent: 4 Demographic characteristics questions, 1 tuning question and 4 target question. Better specified in lines 248-249.

  1. The equations are not numbered.

Thanks, modified as suggested.
5. Verify the following statements: “burnin iterations…” (line 225), it should be burn-in,

Modified as suggested.

 “In our NSU formulation…” (line 208),

 “A sample size of 2000 subjects guarantees a CI of 0.044% .. . ”(Line 254).

Thanks, modified in the text.

  1. The discussion focuses on other aspects rather than the proposed method. This section needs to be improved.

Thanks for pointing this out, modified the discussion as suggested.

  1. Briefly mention what the next stages of your project would consist of.

The next step of this project is to apply the method to the final dataset of this survey and to evaluate if the estimates retrieved will be reliable to the real results.

Submission Date

Reviewer 3 Report

It is my firm belief that this work must be published due to its necessary results for other researchers in the general area of CoViD-19 dynamics. Subnotification is a serious problem for modelling efforts based on official data. 

The text is very well written. As a reviewer I have been very demanding with the correct use of the english language, besides the ability to communicate relevant scientific contributions.

I have 2 suggestions for the authors:

1) in line 83 the possibility of adding "the": "... will continue until the winter of 2020."; and

2) in line 216: (there is no error, here, just a suggestion in style) the possibility of using "2.3.2 Performance of the modified Maltiel estimators" - this would correspond to considering the name "Maltiel" as an adjective qualifying a noun "estimators".

The absence of other comments, in a mathematical sense is not, in any way due to a hurried examination of the text. On the contrary, my reading was careful,as the subject is of my special interest. Because of this my review is highly positive: there is definitely the need for this paper to be published and widely read.

Author Response

It is my firm belief that this work must be published due to its necessary results for other researchers in the general area of CoViD-19 dynamics. Subnotification is a serious problem for modelling efforts based on official data. 

The text is very well written. As a reviewer I have been very demanding with the correct use of the english language, besides the ability to communicate relevant scientific contributions.

We thank the reviewer for the careful consideration and overall positive judgement given to our work.

I have 2 suggestions for the authors:

1) in line 83 the possibility of adding "the": "... will continue until the winter of 2020.";

Modified as suggested.

and

2) in line 216: (there is no error, here, just a suggestion in style) the possibility of using "2.3.2 Performance of the modified Maltiel estimators" - this would correspond to considering the name "Maltiel" as an adjective qualifying a noun "estimators".

Thanks for the suggestion, modified accordingly.

The absence of other comments, in a mathematical sense is not, in any way due to a hurried examination of the text. On the contrary, my reading was careful,as the subject is of my special interest. Because of this my review is highly positive: there is definitely the need for this paper to be published and widely read.

Submission Date

05 March 2021

Date of this review

17 Apr 2021 15:29:49

Bottom of Form

© 1996-2021 MDPI (Basel, Switzerland) unless otherwise stated

Round 2

Reviewer 1 Report

See attached file

Author Response

Reviewer 1

Comments on the manuscript Using social networks to estimate the number of COVID-19 cases: The INCIDENT (hIddeN CovID-19 casEs Network estimation) study protocol Manuscript ID: ijerph-1153833-v2

The paper has been improved in several ways taking into account many of the reviewer’s remarks and concerns. The answers to the reviewers’ comments are satisfactory. In particular, the main aim of the proposed project has been scaled down from the problem of estimating the proportion of “asymptomatic or mildly symptomatic” Covid-19 cases to the problem of estimating the proportion of “undocumented cases” in an exposed population. This seems a more realistic aim for which the proposed approach might be considered, under suitable conditions.

We appreciate the time and effort that the reviewer has dedicated to providing feedback on our manuscript and are grateful for the insightful comments on and valuable improvements to our paper.

Minor remark: In step (2) of the first version (v1) a key question addressed to the respondent was “how many people he/she knows with flu”. In the revised versions (v2) the question becomes “how many people he/she knows with Covid-19”. Obviously, a usable definition of the target population (undocumented cases) is essential in order to define the estimate. Both the questions mentioned above are too loose (v1) or too restrictive (v2) for this purpose. (They even suggest that the protocol has been modified from v1 to v2!)

In the first version of the manuscript, we wrote “flu” as an example, in the second version we choose the question related to the COVID-19. However, as the reviewer as said, this question alone would have been too restrictive. Howewer, we have another target question which is more general (question 5 in the supplementary material) and include symptoms that are related to COVID-19, but not refers directly to it.

 Suggestion: fully report the key questions 5)–6) of the supplementary material or refer to the supplementary material.

Done

The “two major advantages” of NSUM - mentioned in the answers to the reviewers’ comments- with respect to random sampling do not justify neglecting the underlying assumptions. However, the authors promise that “further evaluation of the behaviour of the NSUM estimators will be carried out during data analysis phase”. We’ll see!

A few typing and language errors are still present

Thanks, modified the text.